# The Discovery of Small Allosteric and Active Site Inhibitors of the SARS-CoV-2 Main Protease via Structure-Based Virtual Screening and Biological Evaluation

**DOI:** 10.3390/molecules27196710

**Published:** 2022-10-09

**Authors:** Radwa E. Mahgoub, Feda E. Mohamed, Lara Alzyoud, Bassam R. Ali, Juliana Ferreira, Wael M. Rabeh, Shaikha S. AlNeyadi, Noor Atatreh, Mohammad A. Ghattas

**Affiliations:** 1College of Pharmacy, Al Ain University, Abu Dhabi 64141, United Arab Emirates; 2AAU Health and Biomedical Research Center, Al Ain University, Abu Dhabi 64141, United Arab Emirates; 3Department of Genetics and Genomics, College of Medicine and Health Sciences, United Arab Emirates University, Al-Ain 15551, United Arab Emirates; 4Zayed Centre for Health Sciences, United Arab Emirates University, Al-Ain 15551, United Arab Emirates; 5Science Division, New York University Abu Dhabi, Abu Dhabi 129188, United Arab Emirates; 6Department of Chemistry, College of Science, United Arab Emirates University, Al-Ain 15551, United Arab Emirates

**Keywords:** SARS-CoV-2, M^pro^, structure-based virtual screening, docking, aryl nitrile, allosteric inhibitor

## Abstract

The main protease enzyme (M^pro^) of SARS-CoV-2 is one of the most promising targets for COVID-19 treatment. Accordingly, in this work, a structure-based virtual screening of 3.8 million ligand libraries was carried out. After rigorous filtering, docking, and post screening assessments, 78 compounds were selected for biological evaluation, 3 of which showed promising inhibition of the M^pro^ enzyme. The obtained hits (CB03, GR04, and GR20) had reasonable potencies with K_i_ values in the medium to high micromolar range. Interestingly, while our most potent hit, GR20, was suggested to act via a reversible covalent mechanism, GR04 was confirmed as a noncompetitive inhibitor that seems to be one of a kind when compared to the other allosteric inhibitors discovered so far. Moreover, all three compounds have small sizes (~300 Da) with interesting fittings in their relevant binding sites, and they possess lead-like characteristics that can introduce them as very attractive candidates for the future development of COVID-19 treatments.

## 1. Introduction

Triggering one of the most cataclysmic events in history due to its highly infectious nature, coronavirus disease 2019 (COVID-19) was declared a pandemic by the World Health Organization (WHO) on the 11 March 2020 [1]. Belonging to the large family of coronaviruses (CoV), severe acute respiratory syndrome coronavirus 2 (SARS-CoV-2) is the causative agent behind the outbreak of the novel COVID-19. SARS-CoV-2 was preceded by the severe acute respiratory syndrome coronavirus and the Middle East respiratory syndrome coronavirus, which caused the SARS and MERS outbreaks in China in 2003 and the Middle East in 2012, respectively [2,3]. However, the rapid worldwide spread of COVID-19 has imposed a serious challenge on global public health, urging researchers in different sectors all around the globe to pursue the worldwide quest of searching for novel antivirals and vaccinations for combating the virus. In fact, as of 10 May 2022, the number of confirmed cases, as reported by the WHO, has exceeded 500 million, including more than 6 million deaths [4].

One of the most promising key targets of SARS-CoV-2, which was identified early on, is the main protease enzyme (M^pro^), also referred to as 3-chymotrypsin-like cysteine protease (3CLpro) [5]. M^pro^ plays a key role in the virus’s replication and transcription by cleaving its 1a/1ab polyproteins at multiple sites to produce 16 mature nonstructural proteins essential for replication [6]. This cleavage occurs exclusively after a glutamine residue in the polypeptide sequences, which is a specificity feature not seen in any known human protease [7]. This, coupled with the highly conserved nature of its active site among other coronaviruses (82% and 50% structural similarities with SARS-CoV and MERS-CoV. respectively), has defined M^pro^ as a viable target in drug design and repurposing studies, especially with the continuous and highly paced emergence of new variants of SARS-CoV-2 [8].

Naturally occurring as a homodimer, SARS-CoV-2 M^pro^ is composed of two promotors, each made up of three domains: I (residues 10–99), II (residues 100–182), and III (residues 198–303) [9]. Domains I and II are responsible for active site formation, while domain III plays a vital role in regulating dimerization [10]. The active site is divided into four main subsites: S1, S2, S4, and S1′ [11]. The S1′ subsite contains the catalytic dyad, made up of the active residue, cysteine 145 and histidine 41 [12].

In this study, we utilized structure-based virtual screening to discover small drug-like compounds that can act as inhibitors of the M^pro^ enzyme. Accordingly, we screened a number of commercially available libraries with around 3.8 million ligands in an attempt to find potential M^pro^ inhibitors with small sizes and lead-like characteristics. The inhibition activity of such compounds was evaluated in-vitro using simple enzyme assays and kinetics studies.

## 2. Results and Discussion

### 2.1. Structure-Based Virtual Screening

Figure 1 illustrates the virtual screening protocol that was followed to screen a huge ligand database of more than 3.8 million ligands against the SARS-CoV-2 M^pro^ enzyme. Four commercially available libraries were filtered based on the drug-likeness rules [13,14], and the resulting datasets were then filtered via a PAINS filter using an online open-access server to minimize the possibility of false positives throughout the in-vitro evaluation process. This effectively reduced our collective library size from 3,824,977 ligands to 2,472,261 ligands that were docked stepwise into the M^pro^ active site using three subsequent stages where precision and accuracy increased at the cost of computational time: GLIDE-HTVS, GLID-SP, and GLID-XP. The top-scoring ligands from each library were visually inspected inside the active site of the protein (M^pro^) to shortlist compounds based on their fitting inside the binding site and their interactions with the surrounding residues. The shortlisted compounds were checked for similarity with known aggregators using the online tool Aggregator Advisor. Compounds that came back with no alerts for known aggregate scaffolds were then re-evaluated using MD simulation and MM-GBSA scoring. A total of 178 compounds from all the screened libraries (i.e., TimTec: 59, ChEMBL: 48, and ChemDiv: 60) were further investigated using MD simulation and MM-GBSA scoring.

### 2.2. MD Simulation and MM-GBSA Scoring

Besides the poor estimation of binding affinities being a major drawback of docking, structure-based VS continues to be biased towards the selection of ligands with high molecular weights, regardless of their fitting and binding modes [15]. Hence, a more accurate scoring method such as MM-GBSA is needed to rescore the docked ligands. Accordingly, we conducted MD simulation for 20 ns on the shortlisted compounds for rescoring with MM-GBSA calculations (a total of 178 compounds from all screened libraries). Compared to docking-associated scoring, MM-GBSA has been reported to have a better correlation between the predicted scores and the actual inhibition constant (K_i_) or IC_50_ [16]. As the MM-GBSA scores for all the compounds were very close to one another and all appeared to be low and favorable scores, we calculated the ligand efficiency using the formula below:(1)$$\mathrm{Ligand}\;\mathrm{Efficiency}\;\left({\mathrm{kcal}\;\mathrm{mol}}^{-1}\right)=\frac{\mathrm{MMGBSA}\;\left({\mathrm{kcal}\;\mathrm{mol}}^{-1}\right)}{\mathrm{Number}\;\mathrm{of}\;\mathrm{Heavy}\;\mathrm{Atoms}}$$

This allowed us to overcome the size-related biases of these scoring methods. Table 1 lists examples of hits selected for the in-vitro evaluation, as they showed the best profiles in terms of ligand efficiency, lead-like and drug-like characteristics, and ADMET profile (as shown below).

### 2.3. ADME Profile of Top Hits

A pharmacokinetics and drug-like property assessment was carried out to help us shortlist compounds for in-vitro evaluation. Table 1 showcases the ADME assessment of our top hits. Interestingly, all four top hits were predicted to have high GI absorption rates. CD06 and GR04 were predicted to have the ability to cross the blood–brain barrier (BBB), unlike CB03 and GR20. However, GR04 was also predicted to be a P-gp substrate, which is a transmembrane efflux pump that is highly expressed at the BBB, preventing the accumulation of this potential substrate in the brain [17], hence protecting the brain from any possible side effects attributed to the compound; this was not the case for CD06. All four compounds were predicted to have potential drug–drug interactions, as they all showed the potential to bind to at least three enzymes from the cytochrome P450 family. Our top hits were successful in fulfilling all the drug-like rules depicted in Table 1, with the exception of CD06. Additionally, unlike the others, CD06 did not fulfill the lead-like characteristics, as both its molecular weight and LogP violate the lead-like boundaries. It was also the only compound to come back with a PAINS alert, as it has a Mannich core in its structure, which is a known PAINS-related structure [18]. To sum up, all assessed hits appear to bear drug-like characteristics and can act as good leads for future development, with the exception of CD06, which was predicted to violate a number of rules, and hence its lead candidacy shall be considered with high caution.

### 2.4. Enzyme Assay and IC_50_ Determination

A total of 78 compounds from four different libraries were tested at 100 μM for their inhibition effect against the SARS-CoV-2 M^pro^ enzyme. Compounds that had an inhibition percentage of >20% were further evaluated to find their IC_50_ values. Among all the tested compounds, the IC_50_ values were identified for four compounds, as shown in Table 2 (the IC_50_ plots are shown in Appendix A). Starting with CD06, which had the second best IC_50_ of 226.2 μM, this compound was not a good choice to consider for lead optimization, considering its high molecular weight, complicated structure, and synthetic feasibility. On the other hand, CB03 and GR04 both have rather simple structures with a large space for derivatization. However, they possess IC_50_ values at the high micromolar level (301 μM and 346 μM, respectively). Interestingly, GR20 showed the best inhibition profile when compared to the previous two compounds, bringing the potency down to the middle micromolar level with an IC_50_ value of 92 μM. The latter compound as well as CB03 and GR04 are all worth further investigation, as they all possess small structures along with lead-like characteristics and good ADMET profiles.

### 2.5. Promiscuity Tests

Hits obtained from screening may turn out to be promiscuous with peculiar properties such as poor specificity, aggregate formation, and finally, poor correlation between structure and activity. Accordingly, ruling out the possibility that our hits may be promiscuous compounds exhibiting activity against our target enzyme via nonspecific mechanisms is a crucial step before considering them as proper leads. A number of tests were employed to ensure that our compounds are real inhibitors. The first test was the pre-incubation test, as promiscuous compounds are known to exhibit a time-dependent inhibition, with a decreased (improved) IC_50_ as a result. A competitive inhibitor, on the other hand, would not exhibit the same behavior. The tested compounds were found to have the same inhibition percentage at 100 μM with and without the pre-incubation step, as showcased in Table 3.

The second test employed involved increasing the enzyme concentration 10-fold (46 μM). A nonspecific inhibitor would be attenuated by the presence of excess enzyme. However, the inhibitory effects of CB03, GR04, and GR20 remained unaffected by the increase in the enzyme concentration (Table 3).

The final test was conducted using the detergent bovine serum albumin (BSA) prepared at 1 mg/mL as part of the 20 mM HEPES (pH 6.5) assay buffer. This test could rule out the possibility of the tested compounds forming aggregates within the enzymatic reaction, as an aggregator’s activity would considerably drop in the presence of a detergent because it is unable to form a colloid in the enzyme assay. The inhibition percentage exhibited by the three tested compounds at 100 μM remained the same in the presence and absence of BSA, suggesting that they are, in fact, true inhibitors (Table 3).

### 2.6. Inhibition Kinetic Study of CB03, GR04, and GR20

In order to understand the mechanism of inhibition adopted by our compounds, inhibition kinetics studies were performed for the three best compounds before considering them as lead compounds for future lead optimization. With this information, we can conclude with some degree of certainty where our compounds’ binding sites are (active site or allosteric site). This will, in turn, shape our derivatization process, as it will be in line with the binding sites size (and other known characteristics), making the proposed derivatives optimized for their proposed binding sites. Accordingly, the substrate and compound concentrations were varied, and the data generated from the kinetic study assays were plotted using the Lineweaver–Burk plot and the GraphPad prism tool to plot and evaluate the inhibition mechanism for each of the examined compounds (Figure 2).

CB03 showed no observable change in the measured V_max_, while the K_m_ value increased with the addition of the inhibitor. These findings indicated a competitive mechanism adopted by CB03 to inhibit the M^pro^ active site, as shown in Figure 2a, with a K_i_ of 255.6 μM (Figure 2b). On the other hand, GR04 was shown to fit the noncompetitive model best since the apparent K_m_ remained the same and the V_max_ value was shown to decrease, as shown in the Lineweaver–Burk plot (Figure 2a). Although GR04 turned out to have a high micromolar K_i_ value (Figure 2b), this interestingly suggests that GR04 inhibits the M^pro^ enzyme via binding to an allosteric site rather than the active site (discussed further in latter sections).

In line with the IC_50_ findings, GR20 attained the best K_i_ value (89 μM) among all tested hits, and in terms of mechanism it was shown to fit the mix inhibition model best, as shown in the Lineweaver–Burk plot (Figure 2a). Having said that, competitive behavior cannot be completely ruled out. In fact, GR20 might act with a different mode of inhibition, as the apparent K_m_ and V_max_ parameters were shown to change, and that could be evidence of a nonconventional mechanism, such as covalent inhibition, where the affinity of the substrate decreases if measured at certain timing of the reaction, while the maximum velocity cannot be reached as many enzyme molecules are permanently damaged [19,20]. This argument can be somewhat justified by the presence of the aryl nitrile functionality on the GR20 structure, which is known for its reactivity towards the cysteine residue (that is present in many proteases) [21,22,23]. In fact, many previous studies suggested that the nitrile group has a varied strength of reactivity, which can lead to forming a thioimidate ester adduct with a varied degree of reversibility. Hence, besides being suggested as an allosteric inhibitor, GR20 also seems to have the potential to bind with the M^pro^ catalytic site with an irreversible mechanism [24]. Both proposed mechanisms were further investigated via computational means, as illustrated in the next sections.

#### 2.6.1. CB03 as a Competitive Inhibitor

Molecular docking study: Guided by the kinetics study, we concluded that CB03 occupies the active site (Figure 3a) and exerts its inhibition via a competitive mechanism. Its binding mode was rather interesting (Figure 3b), as it fits snuggly into the subpockets of M^pro^, placing its cyclopentane side chain deep into the S2 subpocket and forming a strong hydrogen bond interaction with the key residue, GLN189, through its carbonyl [8]. The nitrogen atom of the side chain also forms a strong hydrogen bond with HIS164. The bicyclic region of the small compound spans over the S1′ and S1 regions in the pocket, forming a strong hydrogen bond with another key residue, GLY143, through the heteronitrogen in the ring [8]. The benzene ring fits deep into the S1′ pocket to form a stacking interaction with LEU27. A weak interaction is also formed by the carbon atoms in the side chain and another key residue, CYS145 [8]. All in all, CB03 is a weak inhibitor, yet it can be a promising lead for the future development of clinically useful M^pro^ inhibitors, as it is small in size and seems to have a very satisfactory fitting within the targeted active site.

Pairwise energy decomposition analysis of CB03: Further investigation of the interactions between protein residues and CB03 was conducted using a pairwise energy decomposition analysis to calculate the energetic contributions of pairs of residues and ligands to the binding free energy of the complex. Figure 3b presents the energy decomposition for the key residues in the active site with CB03. The top three residues that made significant contributions to the binding affinity are MET165, GLU166, and GLN189. These residues have been identified as key bioactive residues [25]. MET165 and GLN189 form rather strong van der Waals (vdW) and electrostatic interactions, while GLU166 makes a significant contribution to the binding affinity through polar solvation. HIS41 is another key residue that also considerably contributes to the binding affinity through vdW interactions. Other key residues also contribute to the binding affinity but with weaker interactions, such as GLY143 and HIS163.

CB03 has a quinoxaline scaffold, which has previously been investigated for antitumor and antiviral potential against different targets. In fact, quinoxaline derivatives have gained the attention of medicinal chemists in recent years due to their rather interesting biological properties. Quinoxaline-bearing compounds were found to exhibit activity against DNA and RNA viruses [26]. They have also previously been identified as effective potential therapeutic candidates against other viruses, such as HIV-1 [27]. Accordingly, CB03 provides a promising candidate for future development as an antiviral agent against SARS-CoV-2.

#### 2.6.2. GR04 as a Noncompetitive Inhibitor

Through our virtual screening protocol, we aimed to identify reversible inhibitors of the active site of M^pro^. One of the top candidates, GR04, exhibited an excellent fit in the active site in molecular docking and molecular simulations; nevertheless, the kinetics study indicated that GR04 does not bind to the active site. Hence, it appears that GR04 binds to allosteric sites on the molecular surface of M^pro^.

To determine where GR04 binds, we must examine all allosteric sites on the M^pro^ molecular surface. So far, six cavities have been described in the literature to bind fragments or ligands [28,29]. Some are found in distal areas from the main catalytic pocket, and others have been identified on the dimerization interface, a site vital to enzyme activation [30,31]. Given the number of allosteric sites, it was essential to perform a brief druggability assessment to determine their druggability or likelihood of binding drug-like ligands. Then, the molecular docking of GR04 was performed to understand the in-depth interaction of these compounds with the protease’s allosteric sites. Finally, to confirm the binding mode of GR04, MD simulations were carried out, and the ligand stability within the allosteric site was assessed and analyzed. 

Druggability assessment: Using the SiteMap [32] module, we identified the cocrystallized ligand as the core of the pocket that had to be assessed. Druggability scores (Dscores) were then calculated for each of these pockets; the higher the score, the better the druggability (pockets with a Dscore ≥ 1.0 are considered very druggable, while sites with a Dscore of less than 0.8 are classified as difficult nondrug binding sites) [33]. This type of rating is based on three primary factors: pocket size, hydrophobicity, and enclosure [33].

Table 4 shows that, of the six allosteric sites, one had extremely druggable pockets (site #2), one had a druggable pocket (site #5), and one had poor druggability (site #3). Meanwhile, the three remaining sites could not be detected by SiteMap (sites #1, #4, and #6) due to their shallow nature.

Molecular docking study: Molecular docking was carried out for GR04 in both sites #2 and #5, as their Dscores fell within the druggable range (Figure 4a). The docking of GR04 into site #2 resulted in a docking score of −5.44 kcal mol^−1^ (Table 4), the highest docking energy among all allosteric sites. In terms of binding, GR04 showed good fitting inside site #2 (Figure 4b). Next, GR04 was docked into site #5, located directly at the dimerization site, which yielded a docking score of −4.76 kcal mol^−1^. However, the smaller size and enclosure of the pocket resulted in a clash between the propanol side chain of GR04 and the protein backbone, diminishing the final docking score (Figure 4c). The docking GR04 into the shallow interfaces of sites #1 and #3 resulted in docking scores of −3.44 and −3.72 kcal mol^−1^, respectively, as well as poor fitting into the allosteric binding site. Similarly, sites #4 and #6 are very shallow. Thus, docking GR04 into these sites was not possible. When the druggability, docking score, and fitting of GR04 are compared across all reported sites, it seems sensible to conclude that GR04 is most likely exerting its noncompetitive inhibition on the M^pro^ enzyme through binding to allosteric site #2.

Molecular dynamics study: In order to predict the binding mode of GR04 with higher accuracy, we ran MD simulations for the (GR04 site #2, PDB: 7AGA [28] and GR04 site #5, PDB: 5RFA [29]) ligand–protein complex resulting from docking for a further 200 ns, where RMSD measurements were observed for both the ligand molecule and the protein backbone atoms. Additionally, the resulting MD simulations of GR04 were compared to that of the cocrystallized ligand, AT7519. As shown in Figure 5, both compounds seemed fairly stable in site #2, with RMSD values that are less than 2 Å, and the protein backbone also showed a relatively stable structure (1.0–2.5 Å) throughout the 200 ns MD simulations. Despite its fluctuations, the binding of GR04 at site #2 appeared to be more stable compared to site #5, with average RMSD values of 1.5 Å and 2 Å, respectively (Figure 5a). Moreover, the RMSD plots indicate a more stable binding mode of GR04 at site #2 than site #5, as the latter site’s RMSD curve took around 120 ns to converge compared to only a few nanoseconds at site #2. In terms of binding free energy, GR04 seemed to exhibit greater binding energy at site #5 than site #2, as shown by their respective MM-GBSA scores of −26.57 kcal mol^−1^ and −20.44 kcal mol^−1^. These results were comparable to AT7519, which had an MM-GBSA score of −22.50 kcal mol^−1^ (Figure 5b).

Furthermore, for comparison purposes, both compounds were experimentally tested for their inhibition activity at 200 μM, where GR04, interestingly, showed 61% inhibition compared to only 18% shown by AT7519. Future lead optimization efforts can improve the binding free energy by rigidifying the flexible propanol side chain. Such changes would result in a decrease in the loss of entropy and would allow GR04 to form additional interactions with neighboring residues such as PHE294 (Figure 4).

Pairwise energy decomposition analysis of GR04: As we concluded from the previous analysis, GR04 appears to behave similarly to AT7519 in terms of binding stability and affinity but with higher inhibitory efficacy. The next step was to investigate the interactions between protein residues and GR04 in site #2. Using pairwise energy decomposition analysis, we can calculate the energetic contributions of pairs of residues and ligands to the binding free energy of the complex. Figure 6 depicts the energy decomposition of the key residues in the allosteric binding pocket with GR04. The pairwise interactions of the ligand with key residues such as HIS246, GLN107, PRO108, GLN110, ILE249, and PHE294, LYS88, and LYS90 made significant contributions to the binding affinity of the complexes, specifically electrostatic and vdW interactions. Interestingly, the ligand interaction diagram of AT7519 shows that it interacts with GLN110, a residue that contributes to the vdW interactions of GR04. This indicates that GR04 binds in a comparable manner to AT7519. Figure 4b shows a snapshot most representative of the pairwise energy distribution of GR04 within allosteric site #2. It is worth noting that the GR04 binding pocket (Figure 4a,b) is near the dimerization interface, and because M^pro^ is active in its dimeric state, interfering with dimerization limits its catalytic activity. The ligand, GR04, fills up the pocket quite nicely, and its ligand interaction diagram shows several interactions, including a hydrogen bond interaction with HIS246 and two arene-H interactions with GLN107 and ILE249. Hence, the antiviral activity exhibited by GR04 might be attributed to the dimerization-based allosteric mechanism.

Although nothing has been reported regarding the antiviral potential of the main scaffold of GR04 (7H-naphtho[1,2,3-de]quinolin-7-one), benzanthrone has been previously investigated as an antiviral agent. In fact, benzanthrone derivatives have been found to be particularly useful against respiratory syncytial virus (RSV) infections [35,36]. Accordingly, it would be interesting to further evaluate GR04 and its derivatives for their cellular antiviral effect against SARS-CoV-2.

#### 2.6.3. GR20 as an Irreversible vs. Noncompetitive Inhibitor

As discussed above, GR20 could exhibit its inhibition via a noncompetitive mechanism, as suggested by the kinetics study, by binding to an allosteric site, or it could also be a catalytic site binder that is reactive due to its aryl nitrile group. Accordingly, GR20 was further studied in these respective sites to further investigate the likelihood of these two scenarios occurring (Figure 7a).

Molecular docking of GR20: GR20 was redocked into all previously discussed allosteric sites, and as shown previously in the GR04 section, the best docking score and fitting occurred in site #2, which happens to have the best Dscore, as shown in Table 4. Figure 7b shows GR20 in allosteric site #2, where the bicyclic portion of the structure fits snuggly into the allosteric site with the phenol group sticking out of the site. A number of interactions with key residues are formed, including an arene–hydrogen interaction with GLN110 and an arene–arene stacking interaction with PHE294. Two other hydrogen bonds could be seen with ASN151 and THR111.

Since we already established that an irreversible mechanism of action cannot be ruled out because of the possible reactivity of the compound due to its aryl nitrile, GR20 was covalently docked into the M^pro^ catalytic site. The M^pro^ active site is divided into four main subsites: S1, S2, S4, and S1′ [8]. The S1′ subsite contains the catalytic dyad, made up of the active residue CYS145 and HIS41 [9]. Accordingly, GR20 should be able to position its potentially reactive aryl nitrile group in the S1′ subsite to be in close proximity with the CYS145 in order to successfully form a covalent bond with the sulfur in the CYS145.

As seen in Figure 8, interestingly, the resulting pose from the covalent docking is superposed to a great degree on the initial pose from the noncovalent classical docking. The bicyclic portion of the compound spans the middle of the active site, with the amino group pointing upwards, enabling it to form an interaction with the MET49 of the S2 subsite. Meanwhile, the three-carbon-membered side chain is inserted into the S4 subsite and the nitrile is positioned into the S1′ subsite. The hydroxyl group is inserted into the S1 subsite, where it forms interactions with key residues. This adds confidence to the possibility of binding to the catalytic pocket, as the compound was oriented in both docking modes in the same manner, with the nitrile portion of the structure placed in the S1′ subsite, where the CYS145 residue is found [37]. As for the interactions formed, a number of noncovalent interactions were found in both docking modes, as shown in Figure 7c and Figure 8, starting with the hydrogen bond formed with key residue GLU166. Additionally, the hydrogen bond formed by the ammonia and the MET49 residue was also present in both docking modes. The hydroxyl formed a hydrogen bond in both the docking modes, once with PHE140 and another time with ASN142 in the noncovalent docking and covalent docking modes, respectively. The interactions formed by the nitrile with ASN142 and GLY143 in the noncovalent binding mode were substituted by the covalent bond with the CYS145 in the covalent docking mode.

Molecular dynamics study and pairwise energy decomposition analysis of GR20: To ensure that the poses generated by both the noncovalent and covalent docking were accurate, an MD simulation for 100 ns was performed on the noncovalent complex of the compound in the active site, as we wanted to ensure the stability of GR20 within the catalytic binding site. As depicted by the RMSD plot in Figure 9, GR20 appeared to be stable throughout the 200 ns simulation, with its RMSD values around 1.5 Å. The protein backbone also showed relative stability throughout the simulation, with its RMSD ranging from 1.5 to 3 Å. The results from the MD trajectories seem to suggest that the poses predicted by docking are quite stable, meaning that the compound has the perfect fitting within the active site, with its reactive warhead pointing towards CYS145, where the covalent bond can be formed with ease.

As for the interactions formed with residues in the active site, Figure 10 depicts the energy decompositions for the key residues in the active site with GR20. MET165, GLU166, HIS41, CYS145, GLY143, HIE163, PHE140, and SER144 are all previously identified key bioactive residues [25] that are among the top 10 residues in our assessment that make significant contributions to the binding affinity of the complex, especially through vdW interactions in the case of MET165, GLU166, and HIS41 and electrostatic interactions in the case of CYS145, GLY143, HIE163, PHE140, and SER144. To sum up, GR20 appears to be a very promising lead candidate for future irreversible M^pro^ inhibitors, although additional work has to be conducted in the future to confirm its exact mechanism.

To the best of our knowledge, the main scaffold of GR20 has not been tested for biological activity against any known targets. Therefore, we believe that this novel ring scaffold is worth testing for further evaluation of its cellular antiviral potential against the M^pro^ of SARS-CoV-2.

## 3. Materials and Methods

### 3.1. Protein Preparation and Grid Generation

The M^pro^ crystal structure was obtained from the protein data bank (PDB: 6LU7 [5]). Following the elimination of all solvent molecules, the MOE protein preparation module was used to inspect the structure for any missing atoms or residues and correct them accordingly [38]. The Protein Preparation Wizard of the Schrödinger modeling suite [39] was then utilized to prepare the protein by adding hydrogen atoms to the protein structure and assigning partial charges to each atom. Finally, the structure was minimized using the OPLS-3e force field.

In the crystal structure of SARS-CoV-2 M^pro^, a receptor grid was constructed at the surrounding of the centroid of the bound ligand. The grid box was increased to 15 Å for the inner box and 26 Å for the outside box, encompassing the entire binding site cavity.

### 3.2. Ligand Filtration and Preparation

Four libraries were initially downloaded for virtual screening: NCI [40], TimTec [41], ChEMBL [42], and ChemDiv [43]. All libraries were filtered according to the drug-like rules of Lipinski’s rule of five [13] and Veber’s rules [14] (hydrogen bond donor ≤ 5, hydrogen bond acceptor ≤ 10, molecular weight ≤ 500, LogP ≤ 5.0, rotatable bond ≤ 10, and polar surface area (PSA) ≤ 140). We then used a pan-assay interference compounds (PAINS) filter to further filter our libraries, using PAINS-Remover, an online open-access server [44]. The filtered databases were prepared using LigPrep, the ligand preparation module in Schrödinger Maestro [45].

### 3.3. Structure-Based Virtual Screening

The filtered libraries containing PAINS-free drug-like small molecules were docked into the M^pro^ active site using the docking module of the Schrödinger suite (grid-based ligand docking with energetics) as part of the virtual screening (VS) protocol. The employed VS workflow consisted of three subsequent docking steps with increasing levels of precision and accuracy: high-throughput virtual screening (HTVS); standard precision (SP); and extra precision (XP). Moving from one step to the other, the top 20% of the docked libraries were redocked using the next higher mode of docking. Visualization of each library was performed to inspect the binding of the best pose of the top-ranked docked compounds, which were evaluated for their fitting in the active site and the interactions they form with the surrounding residues. Accordingly, compounds from each of the screened libraries were selected, and prior to moving to the next step of molecular dynamics (MD) simulations and MM-GBSA scoring, the online tool Aggregator Advisor [46] was used to ensure that the shortlisted compounds were not aggregators, promiscuous, or nonspecific binders.

### 3.4. MD Simulation and MM-GBSA Scoring

Needing a better understating of the ligands’ binding affinities towards the target pocket, MD simulations were conducted on our shortlisted compounds (178 compounds from all screened libraries). This was also useful to further investigate ligands fitting into the target pocket. Accordingly, AMBER 18 software was used in all MD simulation experiments for the chosen compounds. This was initially started by using the ff19SB and generalized amber force field (GAFF) force fields to assign partial charges and bonding parameters to the protein and ligand atoms, respectively. The ligand–protein complex system was then built with AmberTools’ xleap module, neutralized with Na^+^ counter ions, and solvated with a truncated octahedral box of TIP3P water. Subsequently, the pmemd program in the AMBER 18 package was used to minimize the energy of the whole system in two steps, which included restraining all the solute atoms using a force constant of 500 kcal mol^−1^ Å^−2^ during the minimization of the system for 1000 cycles, followed by minimizing the energy of the whole system for 1000 cycles without applying any restraints. The system was then gradually heated via the NVT ensemble from 0 to 300 K over 20 ps using MD simulation with a 10 kcal mol^−1^ Å^−2^ restraint on ligand atoms. For all bonds involving hydrogen atoms, the SHAKE algorithm was employed. Finally, a 20 ns production MD simulation was run under NPT parameters with a system temperature of 300 K and a pressure of 1.01 × 105 Pa. The ligand efficiency was then calculated for all the compounds, and the top-ranking compounds from each library were then ordered from their respective suppliers. Around 78 compounds were ordered for in-vitro evaluation. The MD simulation was continued until 100 ns for compounds that would later show interesting in-vitro results. In certain cases, the MD simulation was extended for a longer period of time, and a pairwise decomposition analysis (idecomp = 4) was carried out using the obtained MD trajectories to determine the key ligand–residue energetic contributors to the complexes’ binding free energy [47].

### 3.5. Covalent Docking of GR20

The covalent docking of GR20 was performed using CovDock of the Schrödinger suite after the noncovalent docking protocol [48]. The previously prepared protein structure of 6LU7 was used for docking, and the CYS145 residue was selected in the workspace as the reactive residue in the active site. Nucleophilic addition to a triple bond was selected as the reaction type. All the previously generated ligands by LigPrep of GR20 were covalently redocked using the thorough pose prediction mode, ensuring that the side-chain flexibility was taken into account.

### 3.6. Expression and Purification of M^pro^

The recombinant M^pro^ gene, encoding the WT enzyme, was introduced into the pET28b (+) bacterial expression vector by GenScript Inc. (Piscataway, NJ, USA). The Hisx6-tagged M^pro^ enzyme was expressed in E. coli BL21-CodonPlus-RIL (Stratagene). The inoculated culture (2–6 L) was grown in Terrific Broth (TB) at 30 °C in the presence of 100 mg/L kanamycin and 50 mg/L chloramphenicol until the A600 reached 0.8. The temperature was then lowered to 15 °C, and expression was induced overnight with 0.5 mM IPTG. The cells were harvested by centrifugation at 12,000× *g* at 4 °C for 10 min in an Avanti J26-XPI centrifuge (Beckman Coulter Inc.) and resuspended in lysis buffer (20 mM pH 7.8 Tris, 150 mM NaCl, 5 mM imidazole, 3 mM βME, and 0.1% protease inhibitor cocktail from Sigma-Aldrich: P8849). The cells were lysed by sonication on ice and centrifuged at 40,000× *g* for 45 min at 4 °C. The supernatant was loaded on a ProBond Nickel-Chelating Resin (Life Technologies) previously equilibrated with binding buffer (20 mM pH 7.5 Tris, 150 mM NaCl, 5 mM imidazole, and 3 mM βME) at 4 °C. The resin was washed with 10 column volumes (cv) of binding buffer, followed by 15 cv of washing buffer (20 mM pH 7.5 Tris, 150 mM NaCl, 25 mM imidazole, and 3 mM βME). The His-tagged M^pro^ enzyme was eluted from the column in 20 mM pH 7.5 Tris, 150 mM NaCl, 300 mM imidazole, and 3 mM βME and collected in 1 mL aliquots. Finally, the Ni column fractions containing M^pro^ were loaded onto a HiLoad Superdex 200 size-exclusion column (GE Healthcare) using an AKTA purifier core system (GE Healthcare). The column was pre-equilibrated with filtration buffer (20 mM pH 7.5 HEPES, 150 mM NaCl, and 0.5 mM TCEP). The final protein was collected and concentrated to ~150 μM based on the Bradford assay, and the sample purity was assessed via SDS–PAGE.

### 3.7. ADME Screening

To aid us in shortlisting compounds for in-vitro evaluation, an assessment of the compounds’ absorption, distribution, metabolism, and excretion (ADME) was carried out using the SwissADME online server [49]. Information about the compounds’ lead-like and drug-like properties were also provided by the server, along with other pharmacokinetic parameters, such as gastrointestinal absorption, blood–brain barrier permeability, and CYP450 inhibition. Alerts regarding PAINS and reactivity were also provided by the comprehensive assessment carried out by the server.

### 3.8. Enzyme Assay and IC_50_ Determination

The peptide substrate used was a 13 amino acid sequence (KTSAVLQ↓SGFRKM) (GL Biochem (Shanghai) Ltd.) with Dabcyl and Edans attached to its N and C terminals, respectively, and the cleavage site indicated by the ↓ after the glutamine residue. The reaction was performed in 20 mM HEPES (pH 6.5), 120 mM NaCl, 0.4 mM EDTA, 4 mM 1, 4-dithio-D-L-threitol (DTT), and 20% glycerol. All screening compounds were solubilized in DMSO.

The initial screening of the selected library compounds was performed in flat-bottom Nunc black 96-well plates (Thermo scientific) using a fluorescence resonance energy transfer (FRET)-based assay to identify potential inhibitors of M^pro^. The compounds were tested at 100 μM, preceded by the addition of a 10 μM substrate with reaction buffer added up to 100 μL. The enzyme was finally added at a concentration of 4.6 μM to initiate the reaction. DMSO was kept at 10% *V*/*V* in all wells to enhance the solubility of the peptide substrate and the catalytic function of the protease [50]. The plate was incubated at 37 °C for 1 h prior to measuring the fluorescence intensity at 355 nM excitation and 538 nM emission. The compound inhibition capacity was calculated as the percentage of untreated enzyme. The assays were performed in duplicates and repeated three times, and the error bars in the presented graphs indicate S.E.M. Graphs were generated via GraphPad prism 9.3 software. Compounds with sizes suitable for lead optimization and inhibition percentages > 20% were chosen for further evaluation. For IC_50_ measurements, the purified M^pro^ enzyme at a 4.6 μM concentration was incubated with ascending concentrations of the selected compounds in HEPES buffer (pH 6.5) prior to performing the enzymatic assay as previously mentioned. The fluorescence intensities were plotted in reference to the untreated enzyme, and the IC_50_ of each compound was calculated via GraphPad prism 9.3 software. In all presented graphs, error bars indicate S.E.M, and all runs were performed in duplicates and repeated three times.

### 3.9. Promiscuity Tests (BSA and Increasing Enzyme Concentration)

Three methods were used to ensure that our compounds were not peculiar in nature. First, the ‘pre-incubation’ test was carried out, which included the same testing conditions described above but included the addition of the enzyme prior to the substrate and pre-incubating it with the compound for 15 min. The second strategy used was increasing the enzyme concentration by 10-fold while keeping the rest of the assay conditions as they were. The last test performed was the use of a detergent in the prepared buffered. Accordingly, 1 mg/1 mL bovine serum albumin (BSA) in 20 mM HEPES (pH 6.5) was prepared and used to retest the investigated compounds.

### 3.10. Kinetic Study

To determine the method of inhibition exhibited by our compounds of interest, an enzyme kinetics study was performed by varying both the substrate and compound concentrations with a 4.6 μM enzyme concentration. A FRET assay was used in 20 mM HEPES (pH 6.5) buffer. Incubation at 37 °C for 1 h was performed prior to measuring the fluorescence intensity at 355 nM excitation and 538 nM emission. The enzyme concentration was kept constant at 4.6 μM, and the range of the substrate concentration was varied (5 μM, 10 μM, 20 μM, and 40 μM). The compound concentration was also varied (100 μM, 200 μM, and 300 μM).

The average of the generated data was analyzed using a computer-fit calculation (Prism 9.3, GraphPad Software). A Michaelis–Menten graph was plotted in order to identify the changes that occurred in the V_max_ and K_m_ values at each inhibitor concentration and thus mathematically determine the most preferable mode of inhibition.

## 4. Conclusions

Employing structure-based virtual screening, four libraries with around 3.8 million ligands were screened against the M^pro^’s active site after a rigorous filtration protocol. Hits from the multistep docking workflow were selected based on their fitting then rescored via MM-GBSA. Accordingly, 78 compounds were shortlisted and ordered from their respective vendors and were screened in-vitro against the M^pro^ enzyme. Hits that had >20% inhibition were further evaluated in-vitro for their promiscuity and mechanism of inhibition. We ended up with four interesting hits, starting with CD06, which was eliminated as a lead candidate due to its unsatisfactory results in the lead-like/pharmacokinetics assessments. The remaining three hits had rather clean pharmacokinetic-related results and had molecular weights around 300 Da, making them good candidates for a lead. After performing the kinetics study, we could clearly conclude that CB03 exhibits its action via a competitive mechanism, with a K_i_ of 255.6 μM. CB03 was found to have a rather remarkable fitting within the catalytic site, filling the most important subsites in the active site despite its small size. On the other hand, GR04 was found to exhibit a noncompetitive inhibition mechanism, with a K_i_ of 298.1 μM, pointing towards a binding site other than the catalytic pocket. After performing a druggability assessment on M^pro^, we concluded that it may bind to allosteric site #2, as it has the best Dscore, docking score, and overall fitting compared to the other investigated allosteric sites. Finally, we found our best hit, GR20, with a K_i_ of 89.02 μM, which showed a mixed inhibition mode. The presence of an aryl nitrile group on its structure (a known reactive functionality with the ability to be reversible as per its degree of reactivity) led us to propose GR20 as a covalent inhibitor. This is further supported by the fact that the poses generated by the noncovalent docking and covalent docking appear to be almost perfectly superposed. However, although we believe that GR20 is a promising lead candidate, further investigation needs to be conducted using other biological assays to gain a clear understanding of its inhibition mechanism, and thus its binding, before proceeding with the lead optimization process. All in all, the hits discovered in this work seem to be interesting, as they show varied types of inhibition along with small sizes and lead-like characteristics. Hence, they can act as a promising starting point for the future development of clinically useful antiviral agents.

## Figures and Tables

**Figure 1 molecules-27-06710-f001:**
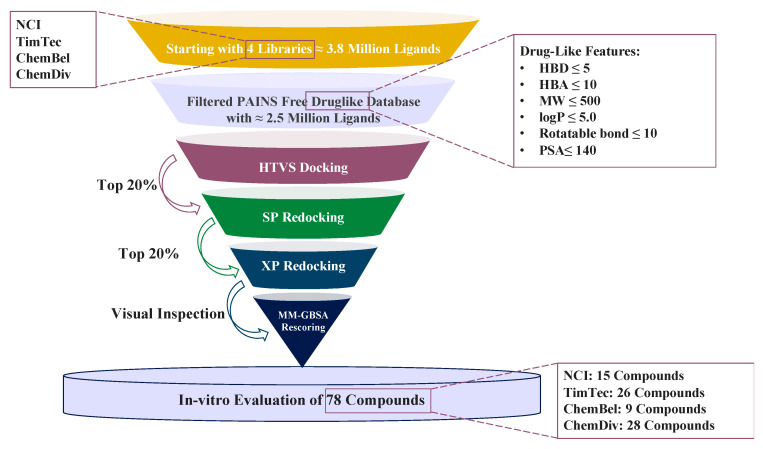
Docking-based virtual screening conducted against the SARS-CoV-2 M^pro^ active site.

**Figure 2 molecules-27-06710-f002:**
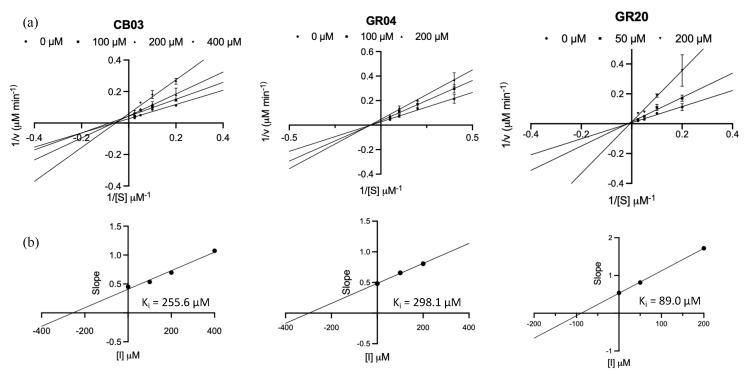
(**a**) The Lineweaver–Burk plot (1/substrate concentration (1/[S]) vs. 1/velocity (1/V)) of CB03, GR04, and GR20, which show competitive, noncompetitive, and mixed inhibition mechanisms, respectively. (**b**) The inhibition constant plots (slope from Lineweaver–Burk linear curves vs. inhibitor concentration [I]) of CB03, GR04, and GR20, where the K_i_ values = −y-intercept.

**Figure 3 molecules-27-06710-f003:**
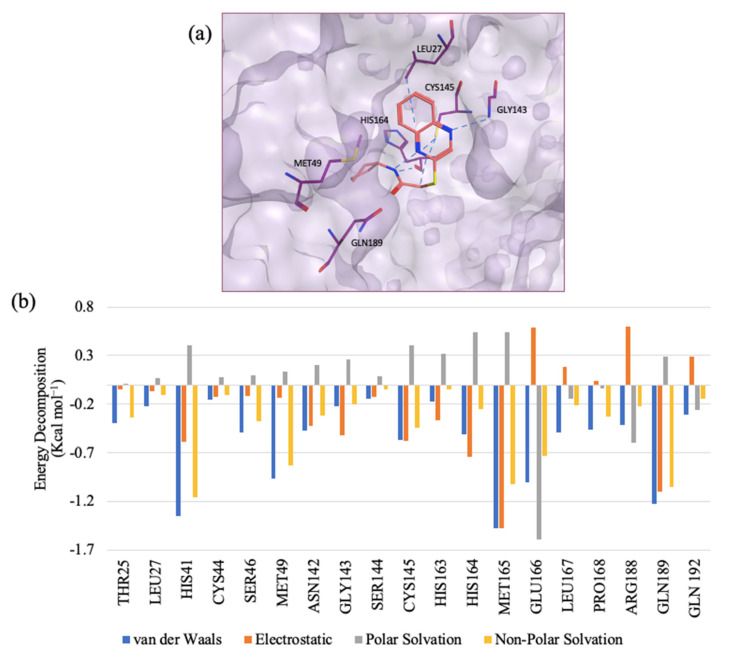
(**a**) The predicted binding mode of CB03 in the target catalytic site. (**b**) Pairwise energy decomposition graph illustrating residue contribution in protein–ligand (CB03) complex.

**Figure 4 molecules-27-06710-f004:**
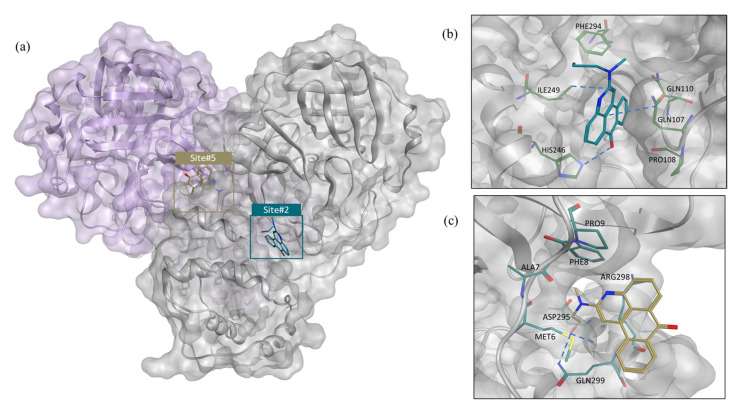
(**a**) The M^pro^ dimer crystal structure showing GR04 as predicted by docking in allosteric sites #2 and #5. (**b**) The predicted binding mode of GR04 in allosteric site #2. (**c**) The predicted binding mode of GR04 in allosteric site #5. For clarity, superimposition on the dimer form of M^pro^ (PDB: 7CAM [34]) was performed to demonstrate binding onto the dimerization site.

**Figure 5 molecules-27-06710-f005:**
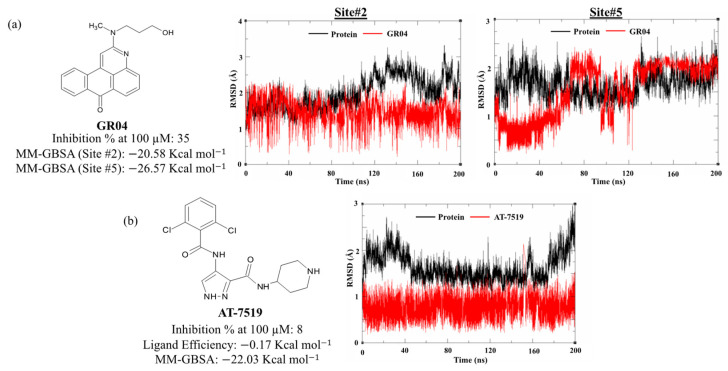
RMSD plot of the 200 ns MD simulations of (**a**) GR04 in sites #2 and #5 and (**b**) AT7519.

**Figure 6 molecules-27-06710-f006:**
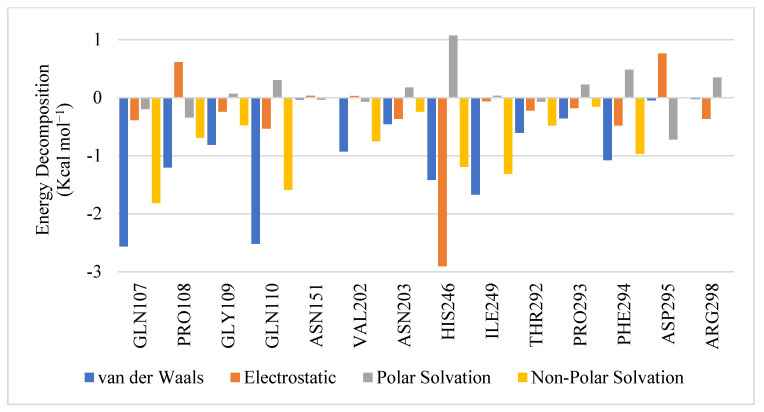
Pairwise energy decomposition graph illustrating residue contributions in site #2 to protein–ligand (GR04) complex.

**Figure 7 molecules-27-06710-f007:**
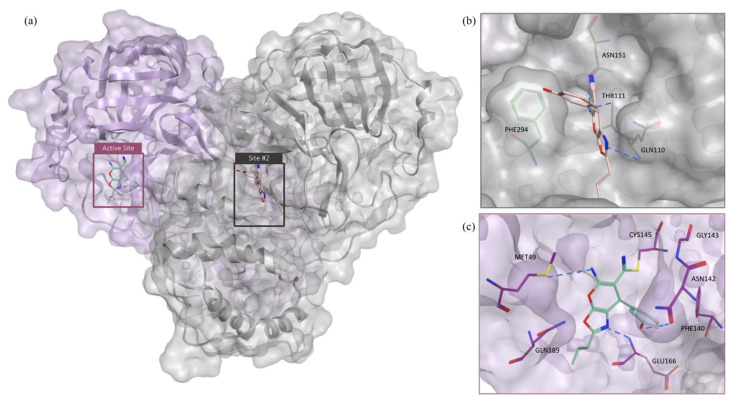
(**a**) Two potential binding sites for GR20, (**b**) GR20 docked in allosteric site #2, and (**c**) GR20 covalently docked in the catalytic site of M^pro^.

**Figure 8 molecules-27-06710-f008:**
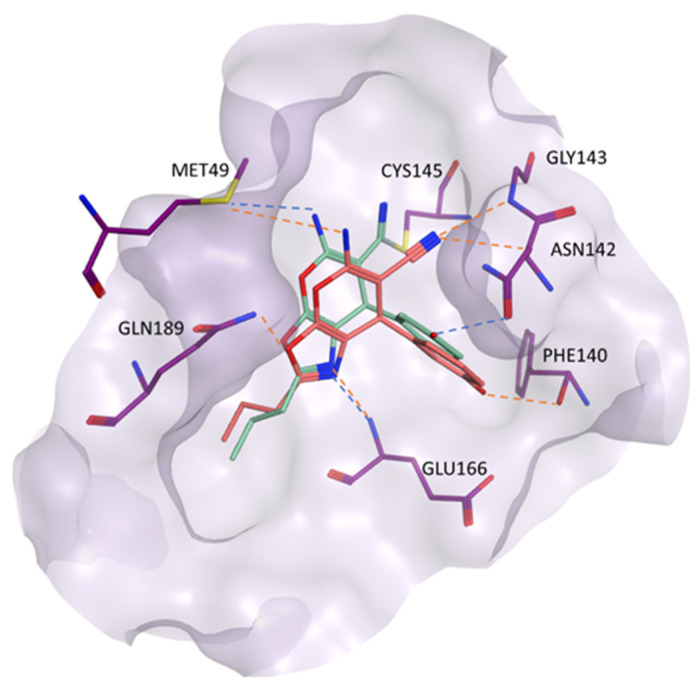
GR20 orientation in M^pro^’s active site with noncovalent docking (pink) and covalent docking (green).

**Figure 9 molecules-27-06710-f009:**
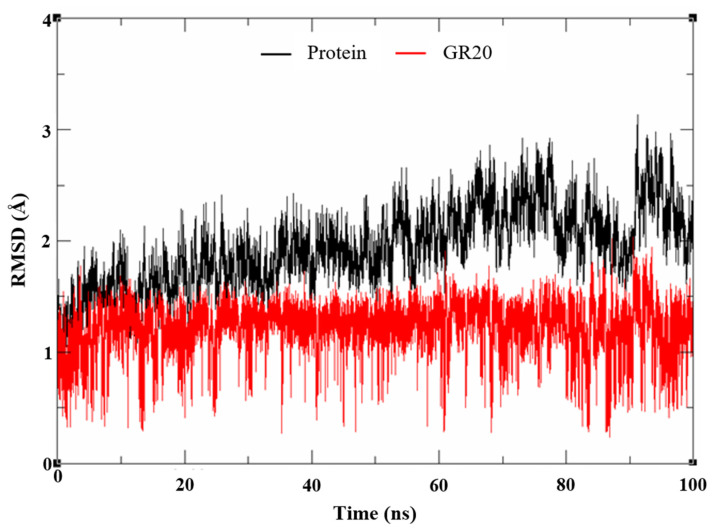
The RMSD plot of 100 ns MD simulations of GR20 in the active site.

**Figure 10 molecules-27-06710-f010:**
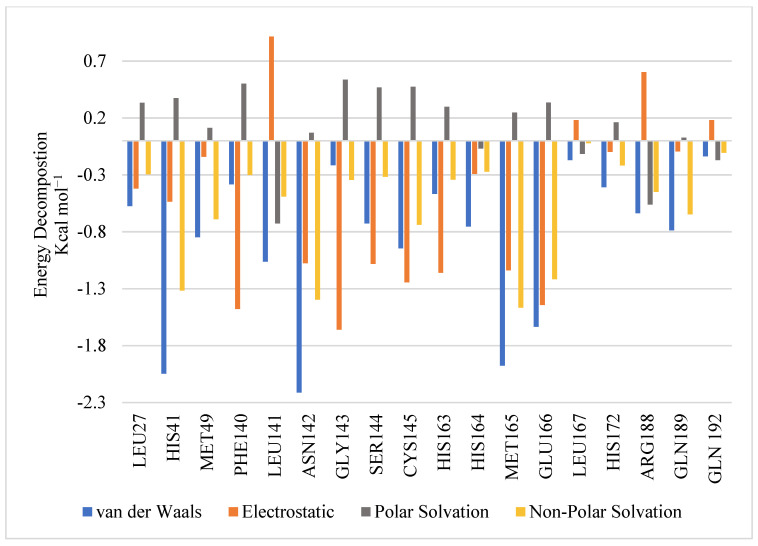
Pairwise energy decomposition graph illustrating residue contributions to protein-ligand (GR20) complex.

**Table 1 molecules-27-06710-t001:** Pharmacokinetics, drug-like, and lead-like characteristics of selected hits predicted by SwissADME.

		CB03	CD06	GR04	GR20
**Pharmacokinetics**	GI absorption	High	High	High	High
	BBB permeant	No	Yes	Yes	No
	P-gp substrate	No	No	Yes	Yes
	CYP1A2 inhibitor	Yes	No	Yes	Yes
	CYP2C19 inhibitor	Yes	Yes	No	Yes
	CYP2C9 inhibitor	Yes	Yes	Yes	Yes
	CYP2D6 inhibitor	Yes	Yes	Yes	No
	CYP3A4 inhibitor	Yes	No	Yes	Yes
	Bioavailability score	0.55	0.55	0.55	0.55
**Drug-likeness**	Lipinski	Yes	Yes	Yes	Yes
	Ghose	Yes	Yes	Yes	Yes
	Veber	Yes	Yes	Yes	Yes
	Egan	Yes	Yes	Yes	Yes
	Muegge	Yes	No	Yes	Yes
			LogP > 3.5		
**Medicinal Chemistry**	PAINS	0	1 alert	0	0
			Mannich in structure		
	Brenk	0	0	0	0
	Lead-likeness	Yes	LogP > 3.5 MW > 350	Yes	Yes

**Table 2 molecules-27-06710-t002:** IC_50_ values of virtual screening top hits.

Compound ID	Structure	Molecular Weight	MM-GBSA Score (kcal mol^−1^)	IC_50_ (μM)
CB03	* 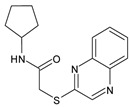 *	304.4	−22.7 ± 3.3	301.0 ± 10.0
GR04	* 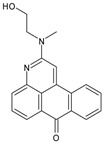 *	287.4	−29.8 ± 3.5	346.0 ± 10.0
GR20	* 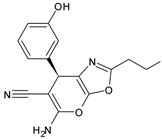 *	297.3	−31.5 ± 4.0	91.8 ± 4.34
CD06	* 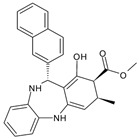 *	412.5	−34.8 ± 5.1	226.2 ± 17.5

**Table 3 molecules-27-06710-t003:** Promiscuity tests results (NT: not tested).

**Pre-Incubation Test**		
	**Inhibition % at 100 μM** **without pre-incubation**	**Inhibition % at 100 μM** **with 15 min pre-incubation**
**CB03**	15.1	14.2
**GR04**	35.4	34.3
**GR20**	NT	NT
**10-Fold Enzyme Concentration Increase**		
	**Inhibition % at 100 μM at** **4.6 μM enzyme concentration**	**Inhibition % at 100 μM at** **46 μM enzyme concentration**
**CB03**	15.0	13.1
**GR04**	33.5	33.9
**GR20**	48.6	48.0
**BSA Buffer**		
	**Inhibition % at 100 μM** **without BSA in assay buffer**	**Inhibition % at 100 μM** **with 1g/mL BSA in assay buffer**
**CB03**	15.5	17.3
**GR04**	33.0	35.0
**GR20**	50.0	51.4

**Table 4 molecules-27-06710-t004:** The Dscore values of the putative M^pro^ allosteric sites along with the docking scores of GR04 obtained for each of the tested pockets (ND: not detected).

Allosteric Site Number	PDB Code	Dscore (Site Map)	GR04 Docking Score (kcal mol^−1^)	GR20 Docking Score (kcal mol^−1^)
Site #1	5REC [29]	ND	−3.76	−3.55
Site #2	7AGA [28]	1.02	−5.31	−5.53
Site #3	7AXM [28]	0.49	−3.68	−2.90
Site #4	5RGJ [29]	ND	ND	−1.94
Site #5	5RFA [29]	0.83	−4.72	−4.90
Site #6	5RF0 [29]	ND	ND	−2.94

## Data Availability

Not applicable.

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
