# Peer review of "The Discovery of Small Allosteric and Active Site Inhibitors of the SARS-CoV-2 Main Protease via Structure-Based Virtual Screening and Biological Evaluation"

_molecules, 2022, doi:10.3390/molecules27196710_

Round 1

Reviewer 1 Report

The manuscript, "The Discovery of Small Leadlike Inhibitors of the SARS-Cov-2 Main Protease via Structure-Based Virtual Screening and Biological Evaluation" describes the discovery of three small molecule ligands with inhibition activity of Mpro enzyme.  There is excellent computational work to show the process of finding thee hits from over 3.8 million molecules that were screened.  There is good discussion of the docking studies where a potential binding site for these inhibitors could be.  This is exciting early results on a significant target.  

One thing that is missing from this manuscript is a more robust method to confirm what the binding affinity of these molecules are for the enzyme.  Preferably confirmed through two methods to determine a Kd for these molecules.  This would strengthen this paper by bostering the idea that the effects that these molecules are having are from binding with the enzyme and not some toxicity effects.

If this information is added to the manuscript I would be excited to see this paper published in this journal. 

Reviewer 2 Report

The work performed by Mahgoub and co-workers aims to explore the inhibition of SARS-CoV-2 Main Protease via structure-based virtual screening and biological evaluation.  The manuscript is overall well-structured. I support its publication considering that only minor points should be addressed.

-          In MM/BSA calculations is not clear how many compounds were selected from each library.

-          The resolution of Figures 4 and 8 should be improved.

-          Had the authors considered running MD simulations for GR04 in site #5? Maybe it could be clarified.

-          Why not consider decomposition energy for CB03 in the target catalytic site? Maybe it should clarify similarities with other competitive inhibitors.

-          In general, the authors could include a brief discussion of the novelties found in this research and how they differ (or agree) with other studies that target this enzyme.

Round 2

Reviewer 1 Report

With the revisions and additional experiments, I would be excited to see this published as is.